# The Eigensharp Property for Unit Graphs Associated with Some Finite Rings

**Heba Adel Abdelkarim [1], Eman Rawshdeh [2,*] and Edris Rawashdeh [3]**

1   Department of Mathematics, Irbid National University, Irbid 21110, Jordan; dr.heba@inu.edu.jo
2   Department of Basic Scientific Sciences, Al-Huson University College, Al-Balqa Applied University, Irbid 21110, Jordan
3   Department of Mathematics, Yarmouk University, Irbid 21110, Jordan; edris@yu.edu.jo
*   Correspondence: eman.rw@bau.edu.jo

**Abstract:** Let $R$ be a commutative ring with unity. The unit graph $G(R)$ is defined such that the vertex set of $G(R)$ is the set of all elements of $R$, and two distinct vertices are adjacent if their sum is a unit in $R$. In this paper, we show that for each prime, $p$, $G(Z_p)$ and $G(Z_{2p})$ are eigensharp graphs. Likewise, we show that the unit graph associated with the ring $Z_p[x]/\langle x^2 \rangle$ is an eigensharp graph.

**Keywords:** commutative ring; unit graph; graph join; biclique; biclique partition number; eigensharp graph

**MSC:** 3A99; 05C25; 05C50; 05C70; 05C76

## 1. Introduction

Studying rings by associating various graphs with the ring via its algebraic structure has attracted the attention many researchers. Beck [1] introduced the zero-divisor graph; Anderson and Badawi [2] introduced the total graph. Grimaldi [3] defined the unit graph $G(Z_n)$ associated with the finite ring $Z_n$, where the author studied some properties of a graph, such as the Hamilton cycles, covering number, independence number, and chromatic polynomial. The units of a ring play a crucial role in determining the structure of the ring, and many features of a ring can be known from these units. So, it is natural to make a connection between a ring with a graph whose edges have a strong relationship with the units of the ring. The unit graph of a ring is one of such graphs.

In 2010, Ashrafi et al. [4] generalized the unit graph $G(Z_n)$ to $G(R)$ for an arbitrary (commutative) ring $R$, and considered standard concepts of graph theory such as connectedness, chromatic index, diameter, girth, and planarity of $G(R)$. Akbari et al. [5] studied the unit graph of a noncommutative ring. Maimani et al. [6] showed that the unit graphs is Hamiltonian if and only if the ring $R$ is generated by its units. Heydari and Nikmehr [7] investigated the case when the ring $R$ is a left Artinian ring. Afkhami and Khosh-Ahang [8] studied the unit graphs of rings of polynomials and power series.

A biclique is a complete bipartite subgraph of $G$. The complete bipartite graphs $K_{1,n}$ are called stars, denoted by $S_n$. A collection $\mathcal{H}_G = \{B_1, B_2, \ldots, B_k\}$ of subgraphs of $G$ is called a biclique partition covering of a graph $G$ if $B_i$ is a biclique subgraph for all $i = 1, 2, \ldots, k$, and for every edge $e \in E(G)$, there exists exactly one $B_i \in \mathcal{H}_G$, such that $e \in E(B_i)$. The biclique partition number of a graph $G$, denoted by $bp(G)$, is given by

$$bp(G) = min\ \{|\mathcal{H}_G| : \mathcal{H}_G \text{and is a biclique partition covering of } G\}.$$

One motivation for studying this parameter is to minimize storage space; listing the subgraphs in a minimum complete bipartite decomposition of $G$ never takes more space than the adjacency list representation. Moreover, the biclique partition number has applications in diverse fields of applied science, such as computational complexity,

automata and language theories, partial orders, artificial intelligence, and geometry (see, for example, [9–13]). When Graham and Pollak [14] first studied this parameter for the complete graph, they were motivated by a network addressing problem. For more details about graph addressing, please see [15]. The adjacency matrix of $G$, denoted by $A(G)$, is a square matrix of order $|V(G)|$, with the $ij$th entry equaling 1 if $v_i v_j$ is an edge of $G$ and 0 otherwise. Witsenhausen (see, for example, [14]) showed that for a graph $G$

$$\max\{a_+(G), a_-(G)\} \leq bp(G),$$

where $a_+(G)$ and $a_-(G)$ are the number of positive and negative eigenvalues of the adjacency matrix $A(G)$, respectively. We repeatedly use this fact below. We say that $G$ is an eigensharp graph if $bp(G) = \max\{a_+(G), a_-(G)\}$, and it is almost eigensharp if $bp(G) = \max\{a_+(G), a_-(G)\} + 1$. Certain families of graphs, including complete graphs $K_n$, complete bipartite graphs $K_{n,m}$, trees, cycles $C_n$ with $n = 4$ or $n \neq 4k$, and various graph products, are eigensharp (see, for example, [16–19]).

The unit graph $G(R)$ is defined such that the vertex set of $G(R)$ is the set of all elements of the ring $R$, and two distinct vertices are adjacent if their sum is a unit in $R$. In this paper, we show that for each prime $p$, $G(Z_p)$, $G(Z_{2p})$ and $G\left(Z_p[x] \big/ \langle x^2 \rangle\right)$ are eigensharp graphs.

## 2. Preliminaries

In this paper, $R$ is assumed to be a commutative ring with unity. An element $a$ is said to be a unit in $R$ if $a$ has a multiplicative inverse. The set $U(R)$ is defined to be the set of all units in $R$. Moreover, the polynomial ring over $Z_n$ is denoted by $Z_n[x]$. In particular, $a$ is a unit in $Z_n$ if the greatest common divisor between $n$ and $a$ is equal to 1. For example, $U(Z_5) = \{1, 2, 3, 4\}$ and $U(Z_6) = \{1, 5\}$.

Several properties of the unit graph are provided in [4], from which we cite the following Theorem:

**Theorem 1.** *[4] Let $R$ be a finite ring. If $2 \in U(R)$, then for every $x \in U(R)$, degree $(x) = |U(R)| - 1$ and for every $x \in R - U(R)$, degree $(x) = |U(R)|$.*

All graphs in this paper are finite undirected simple graphs. For a graph $G = (V(G), E(G))$, the set $V(G)$ denotes the vertex set of $G$, and $E(G)$ denotes the edge set of $G$. The degree of a vertex in $G$ is defined as the number of edges emanating from the vertex. A graph $G$ is said to be $(n, m)$-semiregular if each vertex in $G$ has a degree $n$ or $m$.

For a simple graph $G$, the adjacency matrix $A(G)$ is a symmetric matrix with real eigenvalues such that the algebraic multiplicity is equal to geometric multiplicity for each eigenvalue. We refer to it as multiplicity. It can be proved that $a_+(G) > 0$ and $a_-(G) > 0$ for any non-null graph $G$.

The multiplicity of an eigenvalue $\lambda_i$ is the number of linearly independent eigenvectors associated with it. If $\lambda_i, 1 \leq i \leq j$ are the distinct eigenvalues of the adjacency matrix $A(G)$ with multiplicity $r_i$, then $\sigma(A(G)) = \begin{pmatrix} \lambda_1 & \lambda_2 & \dots & \lambda_j \\ r_1 & r_2 & \dots & r_j \end{pmatrix}$ is called the *spectrum* of $G$. For example,

$$\sigma(A(K_n)) = \begin{pmatrix} n-1 & -1 \\ 1 & n-1 \end{pmatrix} \quad \text{and} \quad \sigma(A(K_{n,m})) = \begin{pmatrix} \sqrt{nm} & 0 & -\sqrt{nm} \\ 1 & nm-2 & 1 \end{pmatrix}.$$

The join of two graphs $G$ and $H$, denoted by $G \vee H$, is the graph with vertex set $V(G \vee H) = V(G) \cup V(H)$ and $E(G \vee H) = E(G) \cup E(H) \cup \{uv : u \in V(G), v \in V(H)\}$. $G \vee H$ is a complete bipartite graph if both $G$ and $H$ are independent vertices. The following Theorem was proved in [20].

**Theorem 2.** *[20] Suppose that G and H are two regular graphs. Then, $a_-(G \vee H) = a_-(G) + a_-(H) + 1$ and $a_+(G \vee H) = a_+(G) + a_+(H) - 1$. Consequently, if each G and H are eigensharp graphs with $bp(G) = a_-(G)$ and $bp(H) = a_-(H)$, then $G \vee H$ is an eigensharp graph.*

### 3. Unit Graph Associated with Rings $Z_p$ and $Z_{2p}$

In this section, we obtain the biclique partition number of $G(Z_p)$, and we prove that $G(Z_p)$ is an eigensharp graph.

**Theorem 3.** *For each prime p, the graph $G(Z_p)$ is eigensharp.*

**Proof.** If $p = 2$ and 3, then $G(Z_p)$ is isomorphic to $P_2$ and $P_3$, respectively. Hence, $bp(G(Z_2)) = bp(G(Z_3)) = 1$ with

$$\sigma(A(G(Z_2))) = \begin{pmatrix} 1 & -1 \\ 1 & 1 \end{pmatrix} \quad \text{and} \quad \sigma(A(G(Z_3))) = \begin{pmatrix} 0 & \sqrt{2} & -\sqrt{2} \\ 1 & 1 & 1 \end{pmatrix}.$$

Hence, for $p = 2$ or 3, $G(Z_p)$ is an eigensharp graph. Now, for $p \geq 5$, let $V = \{0, 1, \ldots, p-1\}$ and $E = \{e_{r,s} : r + s \in U(Z_p)\}$ be the vertex set and the edge set of $G(Z_p)$, respectively. Because $U(Z_p) = \{1, 2, \ldots, p-1\}$ and $2 \in U(Z_p)$, then $|U(Z_p)| = p - 1$. From Theorem 1, it follows that for every $x \in U(Z_p)$, degree $(x) = p - 2$; for every $x \notin U(Z_p)$, degree $(x) = p - 1$. $i = 0$ is the only vertex that has degree $p - 1$ and, for each $i \in Z_p$ with $i \neq 0$, the degree of $i$ is $p - 2$, where $i + (p - i) = 0 \mod p \notin U(Z_p)$, i.e., $i$ and $p - i$ are nonadjacent. Therefore,

$$|E| = \frac{1}{2}[(p-1) + (p-1)(p-2)] = \frac{1}{2}(p-1)^2.$$

Now, let $H = G(Z_p) - \{0\}$ and $A(H)$ be the adjacency matrix of $H$. Then, $G(Z_p)$ is isomorphic to $K_1 \vee H$, where 0 is adjacent to each nonzero element in $Z_p$, and $H$ is a $(p-3)$-regular graph. It has been found and from several computations for different $p$'s that $A(H)$ is a $(p-1) \times (p-1)$ matrix that has the form

$$A(H) = \begin{bmatrix} 0 & 1 & 1 & \cdots & \cdots & 1 & 1 & 0 \\ 1 & 0 & 1 & \cdots & \cdots & 1 & 0 & 1 \\ \vdots & \vdots & \ddots & \cdots & \cdots & \cdot^{\cdot^{\cdot}} & \vdots & \vdots \\ \vdots & \vdots & \cdots & \ddots & \cdot^{\cdot^{\cdot}} & \cdots & \vdots & \vdots \\ \vdots & \vdots & \cdots & \cdot^{\cdot^{\cdot}} & \ddots & \cdots & \vdots & \vdots \\ \vdots & \vdots & \cdot^{\cdot^{\cdot}} & \cdots & \cdots & \ddots & \vdots & \vdots \\ 1 & 0 & 1 & \cdots & \cdots & 1 & 0 & 1 \\ 0 & 1 & 1 & \cdots & \cdots & 1 & 1 & 0 \end{bmatrix}.$$

The enteries of $A(H)$ are all 1, except 0 on the main and secondary diagonals. Notably, the first $\frac{p-1}{2}$ columns are linearly independent. The $\frac{p+1}{2}$th column is the same as the $\frac{p-1}{2}$th column. The $\frac{p+3}{2}$th column is the same as the $\frac{p-3}{2}$th column, $\ldots$, the last column is the same as the first column. Thus, the column rank is $\frac{p-1}{2} = \lfloor \frac{p}{2} \rfloor$. We show that $H$ is eigensharp graph with $bp(H) = a_-(H)$.

Because nullity $(A(H)) = \lfloor \frac{p}{2} \rfloor$, then $\lambda = 0$ is an eigenvalue of $A(H)$ with multiplicity $\lfloor \frac{p}{2} \rfloor$. We notice that the vector $D^{(r)}$, where $r = 2, 3, \ldots, \lfloor \frac{p}{2} \rfloor$ is defined as a $(p-1) \times 1$ vector, and all entries are 0 except the first and last entries, which are 1; the $r$th and $(p-r)$th entries are $-1$, which is an eigenvector for $A(H)$ with eigenvalue $\lambda = -2$. Moreover, because $trace(A(H)) = 0$, then the value $(p-3)$ is an eigenvalue of $A(H)$ of multiplicity 1. Hence,

$$\sigma(A(H)) = \begin{pmatrix} 0 & -2 & p-3 \\ \lfloor \frac{p}{2} \rfloor & \lfloor \frac{p}{2} \rfloor - 1 & 1 \end{pmatrix}.$$

Therefore, $a_-(H) = \lfloor \frac{p}{2} \rfloor - 1 \geq a_+(H)$, and so $bp(H) \geq \lfloor \frac{p}{2} \rfloor - 1$.

Let $\mathcal{H}_H = \{B_i(X_i, Y_i) : 1 \leq i \leq \lfloor \frac{p}{2} \rfloor - 1\}$ be a collection of subgraphs of $H$ such that, for each $i$, $X_i = \{i, p - i\}$ and $Y_i = \{i + 1, i + 2, \ldots, (p - i) - 1\}$ and

$$E(B_i) = \{e_{i,j}, e_{p-i,j} : i + 1 \leq j \leq (p - i) - 1\}.$$

For each $j : 1 \leq j \leq (p - 2i) - 1$, $i + j = 0 \mod p$ only if $j = p - i$, which is completely impossible. Similarly, $(p - i) + (i + j) \neq 0 \mod p$. So, $E(B_i)$ is a nonempty set.

Hence, $B_i$ is isomorphic to $K_{2,(p-1)-2i}$. Note that no pair of edges of $H$ belongs to a common $B_i(X_i, Y_i)$, and

$$\sum_{i=1}^{\lfloor \frac{p}{2} \rfloor - 1} |E(B_i)| = \sum_{i=1}^{\lfloor \frac{p}{2} \rfloor - 1} 2((p - 1) - 2i) = \frac{1}{2}(p - 1)(p - 3) = |E(H)|.$$

Thus, $\mathcal{H}_H = \{B_i(X_i, Y_i) : 1 \leq i \leq \lfloor \frac{p}{2} \rfloor - 1\}$ is a biclique partition of $H$ with cardinality $\lfloor \frac{p}{2} \rfloor - 1$, which implies that $G(Z_p)$ is an eigensharp graph.　□

Now, we show that $G(Z_{2p})$ is an eigensharp graph.

**Remark 1.** *If* $M = \begin{bmatrix} A & B \\ C & D \end{bmatrix}$, *where* $A, B, C$, *and* $D$ *are block matrices, and if* $CD = DC$, *then*

$$\det(M) = \det(AD - BC).$$

*See [21], Theorem 3.*

**Theorem 4.** *The graph* $G(Z_{2p})$ *is eigensharp.*

**Proof.** Note that the graph $G(Z_{2p})$ is a graph with $2p$ vertices. Suppose that the vertex set is $V(G(Z_{2p})) = \{0, 1, 2, \ldots, 2p - 1\}$. Then, the two distance vertices in $G(Z_{2p})$ are adjacent if their sum is an odd number less than $2p$ and not equal to $p$.

Now, the adjacency matrix of $A(G(Z_{2p})) = \begin{bmatrix} 0 & A(K_p) \\ A(K_p) & 0 \end{bmatrix}$ where $A(K_p)$ is the adjacency matrix of the complete graph $K_p$. Using Remark 1, we claim that the spectrum of $\sigma(A(G(Z_{2p}))) = \begin{pmatrix} p - 1 & 1 - p & -1 & 1 \\ 1 & 1 & p - 1 & p - 1 \end{pmatrix}$. To prove this claim, we notice that $\det(\lambda I - A(G(Z_p))) = \det(\lambda^2 I - A^2(K_p)) = \sigma(A(K_p))\sigma(-A(K_p)) = \begin{pmatrix} p - 1 & 1 - p & -1 & 1 \\ 1 & 1 & p - 1 & p - 1 \end{pmatrix}$. So, $bp(G(Z_{2p})) \geq p$. On the other hand, let $\mathcal{H}_{G(Z_{2p})} = \{S_{2k} : 0 \leq k \leq p - 1\}$ be the set of $p$ disjoint stars in $G(Z_{2p})$ generated by the vertices $2k, 0 \leq k \leq p - 1$. Then, $\mathcal{H}_{G(Z_{2p})}$ is a biclique partition of cardinality $p$. Hence, the graph $G(Z_{2p})$ is eigensharp.　□

## 4. Unit Graph Associated with the Ring $Z_p[x] / \langle x^2 \rangle$

In this section, we consider the ring $Z_p[x] / \langle x^2 \rangle = \{a + bX : a, b \in Z_p, X = x + \langle x^2 \rangle\}$, where $\langle x^2 \rangle = \{x^2 P(x) : P(x) \in Z_n[x]\}$ is the ideal of $Z_n[x]$ generated by $x^2$. We show that the unit graph $G(Z_p[x] / \langle x^2 \rangle)$ is eigensharp. We denote the graph $G(Z_p[x] / \langle x^2 \rangle)$ by $G_p(x^2)$.

Let $s = p^2 - p$ and $J_p$ be a $p \times p$ matrix, where all entries are ones; let $1_p$ be a $p \times 1$ matrix, where all entries are ones, $N_p$ be the zero matrix of size $p \times p$, and $0_p$ be the zero matrix of size $p \times 1$. For $m = 1, 2, \ldots, \frac{p-1}{2}$ define the partition matrix $F^{(m)}$ as the $s \times 1$ matrix such that all the submatrices entries are $0_p$, except for the $m$th row, which is the submatrix $1_p$, and the $(p - m)$ row is the submatrix $-1_p$. Furthermore, for $r = 2, 3, \ldots, \frac{p-1}{2}$ defines the partition matrix $H^{(r)}$ as the $s \times 1$, where all the submatrices are $0_p$, except the

first and last rows are the submatrix $1_p$, and the $r$th and $(p - r)$ rows are the submatrix $-1_p$. For example, if $p = 11$, then,

$$
F^{(4)} = \begin{bmatrix} 0_{11} \\ 0_{11} \\ 0_{11} \\ 1_{11} \\ 0_{11} \\ 0_{11} \\ -1_{11} \\ 0_{11} \\ 0_{11} \\ 0_{11} \end{bmatrix}_{110 \times 1} \quad \text{and} \quad H^{(4)} = \begin{bmatrix} 1_{11} \\ 0_{11} \\ 0_{11} \\ -1_{11} \\ 0_{11} \\ 0_{11} \\ -1_{11} \\ 0_{11} \\ 0_{11} \\ 1_{11} \end{bmatrix}_{110 \times 1}
$$

**Theorem 5.** *For each prime $p$, $G_p(x^2)$ is an eigensharp graph.*

**Proof.** Let $a + bX \in {}^{Z_p[x]} \big/ {}_{\langle x^2 \rangle}$. Then, $a + bX$ is a unit if and only if $a$ is a unit in $Z_p$. Thus,

$$
U({}^{Z_p[x]} \big/ {}_{\langle x^2 \rangle}) = \{ r + sX : r, s \in Z_p, r \neq 0 \},
$$

hence, $|U({}^{Z_p[x]} \big/ {}_{\langle x^2 \rangle})| = p(p - 1)$. Because $2 \in U({}^{Z_p[x]} \big/ {}_{\langle x^2 \rangle})$, then, by Theorem 1, $G_p(x^2)$ is a $(p(p-1), p(p-1) - 1)$-semiregular graph.

$T = \{0, X, 2X, \ldots, (p-1)X\}$ is an independent set of $G_p(x^2)$ with each vertex of $T$ having a degree $p(p-1)$. For $v = a + bX \notin T$ and $u = t + sX \in V(G_p(x^2))$, such that $v \neq u$ and $t \in Z_p \backslash \{p - a\}$, we have $v + u \in U(G_p(x^2))$. Thus, $v$ is adjacent with each vertex in $G_p(x^2)$, except $\{a + bX, (p - a), (p - a) + X, \ldots, (p - a) + (p - 1)X\}$, i.e., $v$ has a degree $p^2 - (p + 1) = p(p - 1) - 1$.

Now, we consider the subgraph $W$ of $G_p(x^2)$ induced by $V(W) = V(G_p(x^2)) \backslash T$. Let $m = (p(p - 1) - p - 1)$. Then, $W$ is an $m$-regular graph with

$$
|E(W)| = \frac{1}{2}(p(p - 1) - 1 - p)(p(p - 1)) = \frac{1}{2}[p^4 - 3p^3 + p^2 + p].
$$

It is clear that $G_p(x^2)$ is isomorphic to $T \vee W$. Mainly, we show that $W$ is an eigensharp graph with $bp(W) = a_-(W)$ and, by Theorem 2, $G_p(x^2)$ is an eigensharp.

The adjacency matrix of $W$ is

$$
A(W) = \begin{bmatrix} A(K_p) & J_p & J_p & \cdots & \cdots & J_p & J_p & N_p \\ J_p & A(K_p) & J_p & \cdots & \cdots & J_p & N_p & J_p \\ \vdots & \vdots & \ddots & \cdots & \cdots & \cdot\cdot\cdot & \vdots & \vdots \\ \vdots & \vdots & \cdots & \ddots & \cdot\cdot\cdot & \cdots & \vdots & \vdots \\ \vdots & \vdots & \cdots & \cdot\cdot\cdot & \ddots & \cdots & \vdots & \vdots \\ \vdots & \vdots & \cdot\cdot\cdot & \cdots & \cdots & \ddots & \vdots & \vdots \\ J_p & N_p & J_p & \cdots & \cdots & J_p & A(K_p) & J_p \\ N_p & J_p & J_p & \cdots & \cdots & J_p & J_p & A(K_p) \end{bmatrix}.
$$

Now, we show that

$$
\sigma(A(W))) = \begin{pmatrix} p^2 - 2p - 1 & p - 1 & -(p + 1) & -1 \\ 1 & \frac{p-1}{2} & \frac{p-3}{2} & (p - 1)^2 \end{pmatrix}.
$$

First, because each row of $A(W)$ has $p^2 - 2p - 1$ ones entries, then $A(W)1_s = (p^2 - 2p - 1)1_s$.

Second, because $\lambda = -1$ is an eigenvalue of $A(K_p)$ of multiplicity $p - 1$, then, it is clear that $\lambda = -1$ is an eigenvalue of $A(W)$ of multiplicity $(p - 1)^2$.

Third, if we look to the submatrix in the $(j, 1)$ entry of $A(W)F^{(m)}$, we obtain

$$\left( A(W)F^{(m)} \right)_{j,1} = \begin{cases} 0_p, & \text{if } j \notin \{m, p - m\}, \\ (p - 1)1_p, & \text{if } j = m \\ -(p - 1)1_p, & \text{if } j = p - m. \end{cases}$$

where $j = 1, 2, \ldots, p - 1$ and $m = 1, 2, \ldots, \frac{p-1}{2}$. Thus, $F = \{F^{(m)} : m = 1, 2, \ldots, \frac{p-1}{2}\}$ is a set of linearly independent eigenvectors of $A(W)$ corresponding to the eigenvalue $\lambda = p - 1$.

Fourth, similar to the third case, the $(j, 1)$ entry of $A(W)H^{(r)}$ is

$$\left( A(W)H^{(r)} \right)_{j,1} = \begin{cases} 0_p, & \text{if } j \notin \{1, m, p - m, p - 1\}, \\ (p + 1)1_p, & \text{if } j \in \{1, p - 1\} \\ (p + 1)1_p, & \text{if } j \in \{p - m, m\}. \end{cases}$$

where $j = 1, 2, \ldots, p - 1$ and $r = 2, 3, \ldots, \frac{p-1}{2}$. Thus, $H = \{H^{(m)} : r = 2, 3, \ldots, \frac{p-1}{2}\}$ is a set of linearly independent eigenvectors of $A(W)$ corresponding to the eigenvalue $\lambda = -(p + 1)$. Therefore, the set

$$Q = \{1_s, F^{(1)}, F^{(2)}, \ldots, F^{(\frac{p-1}{2})}, H^{(2)}, H^{(3)}, \ldots, H^{(\frac{p-1}{2})}\}$$

consists of $p - 1$ linearly independent eigenvectors, and because the multiplicity of $\lambda = -1$ is $(p - 1)^2$, then

$$|Q| + (p - 1)^2 = s = p^2 - p.$$

Hence, we obtain $s$ linearly independent eigenvectors of the matrix $A(W)$, which is of size $s \times s$.

Therefore, the characteristic polynomial of $A(W)$ is

$$P(\lambda) = (\lambda + 1)^{(p-1)^2}(\lambda - p^2 + 2p + 1)(\lambda + p + 1)^{\frac{p-3}{2}}(\lambda - p + 1)^{\frac{p-1}{2}},$$

which gives $\sigma(A(W)) = \begin{pmatrix} p^2 - 2p - 1 & p - 1 & -(p + 1) & -1 \\ 1 & \frac{p-1}{2} & \frac{p-3}{2} & (p - 1)^2 \end{pmatrix}$, thus

$$bp(W) \geq (p - 1)^2 + \frac{p - 3}{2} = \left\lfloor \frac{p}{2} \right\rfloor + (p - 1)^2 - 1.$$

Let $[i] : 1 \leq i \leq \left\lfloor \frac{p}{2} \right\rfloor - 1$ denote the class of vertices

$$\{i, i + X, i + 2X, \ldots, i + (p - 1)X\}.$$

Let $[p - i] = \{p - i, p - i + X, p - i + 2X, \ldots, p - i + (p - 1)X\}$. Define $\wp = [i] \cup [p - i]$, $\ell = \bigcup_{j=1}^{i+1}[i + j]$. Then, $|\wp| = 2p$ and $|\ell| = p - 2i - 1$. Now, define $F_i : 1 \leq i \leq \left\lfloor \frac{p}{2} \right\rfloor - 1$ be a biclique subgraph of $W$, such that

$$V(F_i) = \wp \cup \ell$$

and

$$E(F_i) = \{e_{r,s} : r \in \wp, s \in \ell\}.$$

Then, $F_i$ is isomorphic to $K_{2p,p(p-2i-1)}$ with no pair of edges of $E(W)$, which belongs to a common $F_i$ and

$$\sum_{i=1}^{\lfloor \frac{p}{2} \rfloor - 1} |E(F_i)| = 2p^2 \sum_{i=1}^{\lfloor \frac{p}{2} \rfloor - 1} (p - 2i - 1) = \frac{1}{2}p^2(p-1)(p-3).$$

Moreover, $B_j = \{j + tx : 1 \le j \le p - 1, 0 \le t \le p - 2\}$ is a complete subgraph of $W$. Now, consider the disjoint stars $S_{j+tx}$ in $B_j$ generated by the vertices

$$\{j + tx : 1 \le j \le p - 1, 0 \le t \le p - 2\}.$$

Then,

$$\sum_{j=1}^{p-1} \sum_{t=0}^{p-2} |E(S_{j+tx})| = \sum_{j=1}^{p-1} \binom{p}{2} = \frac{1}{2}p(p-1)^2.$$

and

$$\sum_{i=1}^{\lfloor \frac{p}{2} \rfloor - 1} |E(F_i)| + \sum_{j=1}^{p-1} \sum_{t=0}^{p-2} |E(S_{j+tx})| = \frac{1}{2}p^2(p-1)(p-3) + \frac{1}{2}p(p-1)^2 = |E(W)|,$$

which implies that

$$\mathcal{H}_W = \left\{ F_i, S_{j+tx} : 1 \le i \le \left\lfloor \frac{p}{2} \right\rfloor - 1, 1 \le j \le p - 1, 0 \le t \le p - 2 \right\}$$

is a biclique partition of $W$ with cardinality $\lfloor \frac{p}{2} \rfloor + (p-1)^2 - 1$. Therefore, $W$ is an eigensharp graph with $bp(W) = \lfloor \frac{p}{2} \rfloor + (p-1)^2 - 1$, which implies that $G_p(x^2)$ is an eigensharp graph. $\square$

## 5. Conclusions

In this study, for each prime $p$; we proved that the graphs $G(Z_p), G(Z_{2p})$ and $G\left(Z_p[x] \big/ \langle x^2 \rangle\right)$ are eigensharps. We showed that $G(Z_p)$ is isomorphic to a graph $K_1 \vee H$, where $H$ is a certain subgraph of $G(Z_p)$ and $G\left(Z_p[x] \big/ \langle x^2 \rangle\right)$ is isomorphic to $T \vee W$, where $T$ is a certain independent set of $G\left(Z_p[x] \big/ \langle x^2 \rangle\right)$ and $W$ is a certain subgraph of $G\left(Z_p[x] \big/ \langle x^2 \rangle\right)$. Then, the adjacency matrices for $H$ and $W$ were studied to show that $a_-(H) = bp(H)$ and $a_-(W) = bp(W)$, which yields, by Theorem 2, that both graphs $G(Z_p)$ and $G\left(Z_p[x] \big/ \langle x^2 \rangle\right)$ are eigensharps. The spectrum of the graph $A(G(Z_{2p}))$ was found to demonstrate that $bp(G(Z_{2p})) \ge p$. We also described a biclique partition for $G(Z_{2p})$ with cardinality $p$; we hence concluded that $G(Z_{2p})$ is eigensharp.

Finally, we raise the following question: Does the eigensharp property hold for $Z_{p^n}, Z_{pq}$ and $Z_p[x] \big/ \langle x^n \rangle$? We have attempted several examples to answer this question, but our research is still ongoing.

**Author Contributions:** Conceptualization and methodology H.A.A., E.R. (Eman Rawshdeh) and E.R. (Edris Rawashdeh); validation, H.A.A., E.R. (Eman Rawshdeh); formal analysis, E.R. (Eman Rawshdeh); investigation, H.A.A., E.R. (Eman Rawshdeh) and E.R. (Edris Rawashdeh); writing original draft preparation, H.A.A. and E.R. (Eman Rawshdeh); writing review and editing, E.R. (Eman Rawshdeh). All authors have read and agreed to the published version of the manuscript.

**Funding:** The third author is supported by the Scientific Research and Graduate Studies at Yarmouk University.

**Institutional Review Board Statement:** Not applicable.

**Informed Consent Statement:** Not applicable.

**Data Availability Statement:** Not applicable.

**Conflicts of Interest:** The funders had no role in the design of the study; in the collection, analyses, or interpretation of data; in the writing of the manuscript, or in the decision to publish the results.

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
