# Peer review of "The Eigensharp Property for Unit Graphs Associated with Some Finite Rings"

_axioms, doi:10.3390/axioms11070349_

Round 1

Reviewer 1 Report

Mathematical results have been found and are important, but their importance has not been shown in the article. The article has been written in an ordinary format. The article must be improved.

Author Response

Dear Reviewer,
Thank you very much for your consideration, and we really appreciate the comments 
and have learned a lot. Appropriate changes were made in the revised 
manuscript according to the suggestions of reviewers.

1) The manuscript was edited for proper 
English language, grammar, punctuation, spelling, and overall style by two experts of English language.

2) The introduction has been changed to provide a clear motivation for studying the eigensharp property for unit graphs and show the reader why this paper is interested.

3) A new section called Preliminaries has been added to provide some examples and explain some notations that we need throughout the paper.

4) A conclusion has been added at the end of the manuscript.

Please see the attached files.

Reviewer 2 Report

In this manuscript the authors study a particular property of the unit graphs associated to some particular rings. Unfortunately no clear motivation is provided. The reader doesn't understand why this article is interesting and what new it brings in the theory of graphs associated to rings. Besides the state-of-the-art and preliminaries sections are very poor. The authors use some notions/notations that are nort clearly explained, as degree(x) (that then is used also as deg(x)), stars, all-1 matrix, trac(A) (i assume this is the trace of A ands must be denoted as tr(A)). It is not explained why biclique covering is important and why it is different from other types of coverings.

English used in the manuscript is poor, there are paragraphs (as for example the first one on page 4) that are not very clear.

No conclusive section is included, summarizing the main results and explaining their importance, impact on the exisiting theory. What new elements do they bring in this context?

Apart the paper written by the authors, all the references are pretty old.

Based on these comments, I do not recommend the paper for publication in Axioms.

Author Response

Dear Reviewer,
Thank you very much for your consideration, and we really appreciate the comments 
and have learned a lot. Appropriate changes were made in the revised 
manuscript according to the suggestions of reviewers.

1) The manuscript was edited for proper 
English language, grammar, punctuation, spelling, and overall style by two experts of English language.

2) The introduction has been changed to provide a clear motivation for studying the eigensharp property for unit graphs and show the reader why this paper is interested.

3) A new section called Preliminaries has been added to provide some examples and explain some notations that we need throughout the paper. 

4) A conclusion has been added at the end of the manuscript.

5) Several recent references have been added to the list of references. 

Reviewer 3 Report

Report on the manuscript axioms-1787539
In this article, the authors prove that some graphs associated to finite rings defined starting from the ring of all integers are eigensharp graphs. Thus, the relationships between the algebraic structure of a ring and the associated graph(s) are emphasized. Namely, one proves that for each
prime
?, ?(ℤ?) and ?(ℤ2?) are eigensharp graphs. The authors also prove that for any prime ?, the unit graph associated to the ring ℤ?[?]⁄< ?2 > is an eigensharp graph. In Section 1 (Introduction), some notations, definitions and two known results are reviewed. Section 2 is devoted to the unit graph associated to the rings ℤ? and ℤ2?. In section 3, the unit graph associated to the ring ℤ?[?]⁄< ?2 > is discussed. The paper points out new aspects and brings main contribution in the difficult research area of number theory.
Here are a few minor remarks and suggestions, referring to the presentation of this work.
1) For non-specialists, some basic definitions followed by simple examples would be welcome. This could increase the number of interested readers. For example, the notion of the unit of the commutative ring
?, det6ermining all the units in ℤ5, ℤ6, the notation < ?2 > should be under attention of the authors.
2) P.2, line 8 from the bottom: a reference number is missing (it appears as [?]).
3) P.3, Theorem 3: “…the graph … is an eigensharp” seems to not be correct from the English point of view. It would be better to write “…the graph…
is eigensharp” or “…is an eigensharp graph”.
4) P.5, Theorem 5: the meaning of the notation ??(?2) should precede the statement of the theorem.
5) Section 2 (Methods) and the last Section (Discussion) are missing. Please see the Instructions for Authors and/or the Template of the Journal.

Author Response

Dear Reviewer,
Thank you very much for your consideration, and we really appreciate the comments 
and have learned a lot. Appropriate changes were made in the revised 
manuscript according to the suggestions of reviewers.

T1) This manuscript was edited for proper 
English language, grammar, punctuation, spelling, and overall style by two experts of English language.

2) The introduction has been changed to provide a clear motivation for studying the eigensharp property for unit graphs and show the reader why this paper is interested.

3) A new section called Preliminaries has been added to provide some examples and explain some notations that we need throughout the paper.

4) Remarks 2, 3, and 4 that mentioned in the report have been corrected.

Round 2

Reviewer 2 Report

The revised version takes into consideration my comments from the firts report and the paper improved a lot. I suggest its acceptance.